# Hematological malignancy burden in mainland China and Taiwan from 1990 to 2021 and decadal projections: Insights from the global burden of disease study 2021

Yu Fu[1,2], Ya-Zhe Du[1,2], Yun-Wei Zhang[1,2], Fei Song[1,2], Su-Jun Gao[1,2]*, Long Su[1,2]*

**1** Department of Hematology, The First Hospital of Jilin University, Changchun, China, **2** Key Laboratory of Hematology Precision Medicine of Jilin Province, The First Hospital of Jilin University, Changchun, China.

* sulong@jlu.edu.cn (LS), sjgao@jlu.edu.cn (SJG)

## Abstract

### Objectives

Hematological malignancies (HMs) pose a severe threat to human health and contribute substantially to the disease burden in mainland China and Taiwan. Therefore, understanding their burden is crucial for informed decision-making and the effective allocation of healthcare resources.

### Methods

This study utilized the latest data from the Global Burden of Disease 2021 study to describe the epidemiological indices of HMs in mainland China and Taiwan from 1990 to 2021. The future disease burden was projected for the next decade using the Bayesian age-period cohort (BAPC) model.

### Results

Between 1990 and 2021, mainland China experienced an increase in the prevalence and incidence of leukemia and lymphoma, while the mortality and disability-adjusted life years (DALYs) for these diseases declined. Conversely, Taiwan witnessed an overall increase in the prevalence, incidence, mortality, and DALYs of leukemia over the same period. Additionally, multiple myeloma (MM), myelodysplastic/myeloproliferative neoplasms, and other hematopoietic neoplasms have shown significant increases in prevalence, incidence, mortality, and DALYs in China. While the disease burden of myeloid leukemia decreased in mainland China, that of lymphoid neoplasms (including leukemia, lymphoma, and MM) increased, which was not observed in Taiwan. Predictions from the BAPC model suggest that the incidence of several lymphoid neoplasms and MM is expected to increase in mainland China and Taiwan.

**Data availability statement:** All relevant data are within the paper and its Supporting Information files.

**Funding:** This work was supported by the Norman Bethune Program of Jilin University [grant number 2022B17, 2022]; Talent Reserve Program (TRP), the First Hospital of Jilin University [grant number JDYYCB-2023007, 2023]. The funder had no role in study design, data collection and analysis, decision to publish, or preparation of the manuscript. There was no additional external funding received for this study.

**Competing interests:** The authors have declared that no competing interests exist.

## Conclusions

Taiwan continues to face greater challenges in managing HMs compared to mainland China. MM imposes a significant burden on the Chinese population. The findings of this study provide valuable epidemiological insights for optimizing the allocation of medical resources.

## Introduction

Hematological malignancies (HMs), including leukemia, lymphoma, and multiple myeloma (MM), represent a significant global health burden with diverse clinical characteristics, treatment responses, and prognoses. Leukemia, the most common type of blood cancer, encompasses both acute and chronic subtypes with striking disparities in outcomes. Acute lymphoblastic leukemia (ALL) in children achieves cure rates exceeding 90%; nonetheless, it remains a leading cause of pediatric cancer death due to relapses in standard-risk patients [1]. In contrast, adult ALL patients face poor survival with conventional chemotherapy, although targeted therapies such as monoclonal antibodies and tyrosine kinase inhibitors (TKIs) have improved outcomes [2]. Acute myeloid leukemia (AML), the dominant acute leukemia in adults, has shown survival gains with FLT3/IDH inhibitors and Bcl-2 antagonists; however, resistance and relapse limit the 5-year survival to approximately 30% [3,4]. Furthermore, chronic leukemia, such as chronic lymphocytic leukemia (CLL) and chronic myeloid leukemia (CML), is influenced by age and ethnicity. CLL is prevalent in older adult Caucasians and is managed with active surveillance or BTK/Bcl-2 inhibitors [5,6], while the introduction of TKIs in CML reduced mortality from 10%−20% to 1%−2%, though resistance and toxicity remain challenges [7–9]. Lymphomas exhibit complex landscapes. Patients with Hodgkin lymphoma (HL), which affects adolescents and young adults, currently achieve a 5-year overall survival (OS) of approximately 90% with chemotherapy and immunotherapy; however, long-term survivors face treatment-related toxicities [10]. Additionally, non-Hodgkin lymphoma (NHL), a heterogeneous group, ranges from aggressive subtypes requiring intensive (but curative) therapy, which are often associated with cardiovascular complications or secondary cancers [11,12], to indolent forms that persist despite treatment, leading to chronic morbidity [11,13]. CAR T-cell therapy offers hope for relapsed/refractory NHL; however, challenges in accessibility, manufacturing, and toxicity limit its use [14,15], whereas bispecific antibodies have emerged as promising alternatives [14]. MM is the second most common HM and has improved survival with immunomodulatory agents and proteasome inhibitors; nonetheless, it remains incurable, with most patients relapsing or developing resistance [16]. Notably, CAR-T cell therapy and bispecific antibodies are emerging as unproven solutions [17,18].

Against this global backdrop, China, home to nearly one-fifth of the world's population, faces unique epidemiological features of HMs, including relatively younger diagnostic ages and higher frequencies of favorable molecular markers [19,20], which may influence treatment responses and resource allocation. However, the disease

burden of HMs in China, including Taiwan, remains unclear. While prior Global Burden of Disease (GBD) studies analyzed leukemia and lymphoma at global or regional scales [21,22], none focused specifically on China or integrated recent advancements in diagnostic technologies and targeted therapies (e.g., TKIs and CAR-T cells), which may have altered the incidence, mortality, and survival trajectories. A comparative China-United States leukemia study omitted Taiwan and lacked granularity on the therapy-related impacts [23]. This study addresses these gaps by leveraging GBD 2021 data to quantify the burden of HMs in China (1990–2021), identify trends and regional disparities, forecast future trajectories, and contextualize findings within the evolving therapeutic landscape. By linking epidemiological insights to advancements in precision oncology, immunotherapy, and resource allocation, this analysis aimed to provide clinicians and policymakers with evidence to optimize healthcare strategies, particularly for emerging treatments such as CAR-T cell therapy, and address the unique challenges of China in managing these complex diseases.

## Methods

### Data source

Publicly available data on HMs in mainland China and Taiwan (a province of China) were derived from the 2021 GBD investigation on November 20th 2024 (https://www.healthdata.org), which provided a comprehensive evaluation of health impairments using the latest epidemiological data and refined methodologies [24]. These datasets are freely usable for noncommercial research under the Institute for Health Metrics and Evaluation (IHME). The HMs analyzed in this study included ALL, AML, CLL, CML, other leukemia, HL, Burkitt lymphoma (BL), NHL, MM, myelodysplastic, myeloproliferative (MD/MP), and other hematopoietic neoplasms. Here, we extracted values and their 95% uncertainty intervals (UI) for the prevalence, incidence, mortality, and disability-adjusted life years (DALYs) of HMs from the GBD 2021 dataset. Age-standardized prevalence rates (ASPR), incidence rates (ASIR), mortality rates (ASMR), and DALY rates (ASDR) were selected for cancer epidemiological studies using the GBD database, as these metrics comprehensively characterize the disease burden across multiple dimensions. The ASPR captures the population disease burden at a specific time to inform resource allocation, and the ASIR identifies new cases to assess risk profiles and primary prevention effectiveness. The ASMR reflects disease severity and healthcare system impact, whereas the ASDR integrates mortality and disability to provide a holistic view of health losses. Together, these metrics align with the GBD's global standardized framework for comparative analysis. The GBD study ensures data quality through rigorous standardization and statistical processing, including the selection of reliable data sources, dataset cleaning, application of uniform definitions, use of advanced modeling, and multilevel reviews by global collaborators. Given the preexisting data quality control processes for GBD, no additional exclusion criteria were applied in this study. This study relied on publicly available data without identifiable personal information; therefore, it did not require additional ethical approval or informed consent from the patients.

### Estimated annual percentage change and percentage change

The estimated annual percentage change (EAPC) is a reliable and well-established metric. It has been widely used in previous studies to assess trends in ASPR, ASIR, ASMR, and ASDR over a specific period. A regression model calculates the EAPC by treating time as the independent variable and assuming that the natural logarithm of the ASR is linear with time [24]. The calculation formula for EAPC is as follows: $y = \alpha + \beta x + \varepsilon$, where $y$ = the natural logarithm of rates, $\alpha$ = intercept, $x$ = calendar year, $\beta$ = the slope representing the positive or negative trends of rates, and $\varepsilon$ = the random error. EAPC = 100 × (exp ($\beta$) −1), and its 95% confidence intervals (CI) also could be calculated from this model [25].

### Joinpoint regression analysis

A joinpoint regression analysis model was used to evaluate temporal trends in disease prevalence, incidence, mortality, and DALYs, which can identify joinpoints and key years with statistically significant shifts in incidence trends to split

continuous data into segmented phases, outperforming single-trend models for detecting inflection points in disease rates [26]. This model can calculate the annual percent change (APC) and its 95% CI, providing insights into trends over specified time periods. Additionally, the average annual percentage change (AAPC) was calculated to provide a comprehensive overview of the overall trends from 1990 to 2021. An estimated APC or AAPC with a 95% CI lower limit above zero indicated an upward trend in the specified period. Conversely, APC or AAPC with a 95% CI upper limit below zero signifies a downward trend. If the 95% CI for APC or AAPC was zero, the trend was stable. Detailed equations and model formulations for joinpoint regression analysis have been previously described [27].

### Predicting future disease burden of HMs

The Bayesian Age-Period-Cohort (BAPC) model was used to predict the disease burden of HMs from 2022 to 2032. The BAPC model integrates three key dimensions: age effects (changes in risk with age), period effects (temporal trends affecting all age groups during specific time frames), and cohort effects (birth cohort-specific influences such as generational exposures) [28]. It operates under the Bayesian framework using Markov Chain Monte Carlo methods to estimate parameters with prior distributions, which helps reduce overfitting in small sample sizes [28]. This model is particularly suitable for diseases such as HMs, in which trends are influenced by aging populations, evolving treatment landscapes, and environmental factors. Compared to traditional models, BAPC demonstrates superior predictive performance because of its ability to capture complex, nonlinear trends and incorporate prior knowledge, leading to more stable estimates [29]. Moreover, BAPC's flexibility of the BAPC in handling sparse data and providing probabilistic predictions (via posterior distributions) enhances its robustness in projecting long-term trends, making it the preferred choice for comprehensive disease burden forecasting [30]. By combining empirical data with prior insights, this model provides unique parameter estimates that ensure reliable and consistent results. We downloaded the new World Standard Population data (2000–2025) from the World Health Organization via the National Cancer Institute. Furthermore, global population projection data were extracted from the IHME agency, and a BAPC model was established to forecast the future disease burden.

### Statistical analysis

The rates of prevalence, incidence, mortality, and DALYs were reported per 100,000 individuals, including those with 95% of UI. All analytical procedures and graphical depictions were performed using JoinPoint Desktop software in conjunction with the statistical computing software R (Version 4.4.1).

## Results

### Overall disease burden of HMs in China

In mainland China, the number of leukemia cases rose from 197,970 (95% UI: 150,786 − 236,909) in 1990–531,661 (95% UI: 352,729 − 665,154) in 2021, and the corresponding ASPR increased from 18.04 (95% UI: 13.83 − 21.68) per 100,000 individuals to 40.09 (95% UI: 26.30 − 52.58) per 100,000 persons from 1990 to 2021 with an EAPC at 3.23 (95% CI: 3.00 − 3.45) (Table 1). Most leukemia subtypes showed an upward trend in ASPR, particularly ALL with an EAPC of 4.99 (95% CI: 4.59 − 5.40) and CLL with an EAPC of 4.82 (95% CI: 4.61 − 5.03). However, AML exhibited a decline in ASPR from 2.25 (95% UI: 1.13 − 3.76) per 100,000 persons to 1.62 (95% UI: 1.10 − 2.26) per 100,000 individuals (Table 1). The incidence of leukemia reached 105,667 (95% UI: 75,276 − 132,237) in 2021, an increase from 76,204 (95% UI: 58,312 − 90,958) in 1990. During this same period, the ASIR rose from 7.14 (95% UI: 5.52 − 8.58) per 100,000 persons to 7.21 (95% UI: 4.93 − 9.05) per 100,000 population (Table 1). The ASIR of lymphoid leukemia showed an increasing trend, with CLL being the most significant subtype (EAPC = 2.53; 95% CI: 2.43 − 2.64), while that of other leukemia subtypes decreased during this period, and the most significant decrease was noted in CML with an EAPC of −2.33 (95% CI: −2.52 to −2.14) (Table 1). The number of mortality cases and ASMR of leukemia in 2021 were 58,903 (95% UI:

**Table 1. Cases and age-standardized rates of hematological malignancies in mainland China and their temporal trends.**

| | 1990 | | 2021 | | 1990-2021 |
|---|---|---|---|---|---|
| | Cases No. (95% UI) | ASR per 100,000 No. (95% UI) | Cases No. (95% UI) | ASR per 100,000 No. (95% UI) | EAPC (95% CI) |
| **Prevalence** | | | | | |
| Leukemia | 197,970 (150,786 – 236,909) | 18.04 (13.83 – 21.68) | 531,661 (352,729 – 665,154) | 40.09 (26.30 – 52.58) | 3.23 (3.00 – 3.45) |
| AML | 74,668 (52,614 – 100,415) | 6.60 (4.65 – 8.87) | 188,395 (101,859 – 268,279) | 22.02 (11.49 – 32.37) | 4.99 (4.59 – 5.40) |
| ALL | 25,514 (12,615 – 42,911) | 2.25 (1.13 – 3.76) | 24,172 (16,449 – 33,827) | 1.62 (1.10 – 2.26) | −1.54 (−1.80 to −1.28) |
| CLL | 28,080 (16,312 – 38,004) | 2.72 (1.61 – 3.70) | 202,360 (125,446 – 282,424) | 9.84 (6.09 – 13.75) | 4.82 (4.61 – 5.03) |
| CML | 3986 (1803 – 5818) | 0.37 (0.17 – 0.54) | 8386 (4238 – 12,742) | 0.47 (0.23 – 0.71) | 0.63 (0.50 – 0.77) |
| Other leukemia | 65,722 (36,085 – 91,219) | 6.10 (3.46 – 8.44) | 108,349 (52,659 – 152,288) | 6.14 (2.90 – 8.44) | 0.15 (0.08 – 0.22) |
| HL | 12,935 (5607 – 17,906) | 1.16 (0.50 – 1.61) | 23,818 (14,287 – 31,455) | 1.36 (0.83 – 1.79) | 0.50 (0.35 – 0.64) |
| NHL | 80,302 (68,643 – 95,538) | 7.19 (6.19 – 8.55) | 613,152 (476,112 – 762,582) | 31.17 (24.42 – 38.29) | 5.45 (5.19 – 5.72) |
| BL | 2172 (1033 – 3263) | 0.20 (0.10 – 0.30) | 8630 (3933 – 12516) | 0.53 (0.25 – 0.75) | 2.87 (2.61 – 3.13) |
| Other NHL | 78,130 (66,934 – 93,318) | 6.99 (6.02 – 8.37) | 604,522 (469,849 – 750,199) | 30.64 (24.04 – 37.67) | 5.51 (5.23 – 5.79) |
| MM | 2977 (2052 – 5851) | 0.32 (0.22 – 0.64) | 47,004 (29,544 – 62,136) | 2.19 (1.37 – 2.90) | 5.96 (5.35 – 6.56) |
| MD/MP & other HN | 596,754 (484,498 – 725,804) | 62.55 (50.91 – 77.15) | 1,480,301 (1,206,037 – 1,778,158) | 70.54 (58.25 – 83.76) | 0.70 (0.56 – 0.83) |
| **Incidence** | | | | | |
| Leukemia | 76,204 (58,312 – 90,958) | 7.14 (5.52 – 8.58) | 105,667 (75,276 – 132,237) | 7.21 (4.93 – 9.05) | 0.19 (0.06 – 0.32) |
| AML | 38,025 (26,760 – 50,354) | 3.38 (2.39 – 4.49) | 38,571 (21,149 – 50,762) | 3.64 (2.00 – 5.05) | 0.63 (0.34 – 0.92) |
| ALL | 15,309 (8206 – 24,142) | 1.46 (0.80 – 2.24) | 17,835 (11,876 – 24,800) | 1.03 (0.69 – 1.45) | −1.51 (−1.70 to −1.32) |
| CLL | 6780 (3988 – 9157) | 0.72 (0.42 – 0.98) | 28,927 (17,958 – 40,579) | 1.42 (0.88 – 2.00) | 2.53 (2.43 – 2.64) |
| CML | 3915 (1816 – 5747) | 0.38 (0.17 – 0.55) | 3850 (2087 – 6105) | 0.21 (0.11 – 0.33) | −2.33 (−2.52 to −2.14) |
| Other leukemia | 12,174 (6913 – 16,818) | 1.20 (0.70 – 1.66) | 16,484 (8183 – 23,107) | 0.91 (0.44 – 1.26) | −0.81 (−0.86 to −0.77) |
| HL | 4746 (2015 – 6561) | 0.48 (0.20 – 0.67) | 4211 (2542 – 5542) | 0.23 (0.14 – 0.30) | −2.73 (−3.00 to −2.47) |
| NHL | 31,216 (26,893 – 37,951) | 3.32 (2.86 – 4.03) | 110,924 (86,934 – 135,200) | 5.53 (4.36 – 6.68) | 1.88 (1.59 – 2.16) |
| BL | 296 (145 – 444) | 0.03 (0.01 – 0.04) | 1322 (597 – 1920) | 0.08 (0.04 – 0.11) | 3.20 (2.93 – 3.46) |
| Other NHL | 30,920 (26,700 – 37,655) | 3.29 (2.84 – 4.00) | 109,602 (86,005 – 133,918) | 5.44 (4.31 – 6.60) | 1.86 (1.57 – 2.16) |
| MM | 1693 (1154 – 3360) | 0.20 (0.13 – 0.39) | 17,250 (11,017 – 22,663) | 0.81 (0.52 – 1.07) | 4.05 (3.38 – 4.73) |
| MD/MP & other HN | 32,561 (26,326 – 40,365) | 3.20 (2.59 – 3.97) | 80,882 (65,942 – 99,150) | 3.88 (3.21 – 4.69) | 0.99 (0.81 – 1.17) |
| **Mortality** | | | | | |
| Leukemia | 67,423 (52,045 – 80,043) | 6.46 (5.04 – 7.66) | 58,903 (43,626 – 74,039) | 3.42 (2.51 – 4.26) | −2.22 (−2.31 to −2.14) |
| AML | 33,919 (23,963 – 44,728) | 3.05 (2.17 – 4.02) | 20,613 (11,781 – 27,302) | 1.36 (0.78 – 1.75) | −2.70 (−2.79 to −2.61) |
| ALL | 14,853 (8014 – 23,402) | 1.45 (0.80 – 2.21) | 15,311 (10,365 – 21,401) | 0.88 (0.59 – 1.24) | −2.04 (−2.26 to −1.81) |
| CLL | 4854 (2824 – 6573) | 0.55 (0.33 – 0.74) | 8636 (5527 – 12384) | 0.44 (0.28 – 0.63) | −0.80 (−0.90 to −0.70) |
| CML | 3591 (1689 – 5240) | 0.36 (0.17 – 0.52) | 1841 (1080 – 3094) | 0.10 (0.06 – 0.16) | −4.72 (−5.03 to −4.41) |
| Other leukemia | 10,205 (6022 – 13,819) | 1.05 (0.64 – 1.49) | 12,503 (6547 – 17,075) | 0.65 (0.34 – 0.88) | −1.52 (−1.57 to −1.47) |
| HL | 4393 (1869 – 6058) | 0.47 (0.20 – 0.64) | 2443 (1507 – 3232) | 0.13 (0.08 – 0.17) | −4.71 (−4.94 to −4.49) |
| NHL | 24,023 (20,737 – 29,384) | 2.67 (2.31 – 3.27) | 42,857 (33,553 – 51,712) | 2.13 (1.68 – 2.57) | −0.74 (−0.93 to −0.54) |
| BL | 244 (119 – 350) | 0.02 (0.01 – 0.03) | 241 (115 – 339) | 0.01 (0.01 – 0.02) | −2.52 (−2.99 to −2.06) |
| Other NHL | 23,780 (20,539 – 29,147) | 2.65 (2.29 – 3.24) | 42,616 (33,377 – 51,460) | 2.12 (1.67 – 2.56) | −0.72 (−0.92 to −0.53) |
| MM | 1591 (1080 – 3159) | 0.19 (0.13 – 0.39) | 12,984 (8448 – 17,114) | 0.62 (0.40 – 0.81) | 3.11 (2.40 – 3.83) |
| MD/MP & other HN | 1246 (622 – 2627) | 0.16 (0.08 – 0.34) | 4744 (2800 – 8778) | 0.25 (0.15 – 0.45) | 1.68 (1.55 – 1.82) |
| **DALYs** | | | | | |
| Leukemia | 3,924,466 (2,969,934 – 4,726,963) | 343.57 (260.79 – 414.78) | 2,205,221 (1,612,839 – 2,736,625) | 151.54 (108.71 – 185.06) | −2.90 (−3.02 to −2.78) |
| AML | 2,275,291 (1,587,181 – 3,041,612) | 195.79 (136.54 – 262.37) | 924,422 (525,902 – 1,182,320) | 74.06 (42.86 – 94.99) | −3.31 (−3.44 to −3.18) |
| ALL | 851,931 (419,991 – 1,454,750) | 75.09 (37.56 – 127.44) | 548,555 (373,859 – 778,262) | 36.96 (25.32 – 52.46) | −2.81 (−3.07 to −2.55) |
| CLL | 176,963 (101,444 – 240,861) | 17.07 (9.95 – 23.04) | 268,251 (165,772 – 380,358) | 14.11 (8.74 – 19.93) | −0.64 (−0.73 to −0.56) |
| CML | 177,796 (80,345 – 260,190) | 15.59 (7.13 – 22.70) | 60,929 (35,085 – 101,126) | 3.65 (2.06 – 6.03) | −5.49 (−5.83 to −5.15) |
| Other leukemia | 442,485 (246,386 – 612,954) | 40.03 (22.76 – 54.83) | 403,064 (200,882 – 557,663) | 22.76 (11.20 – 31.14) | −1.85 (−1.90 to −1.79) |

*(Continued)*

**Table 1.** (Continued)

| | 1990 | | 2021 | | 1990-2021 |
|---|---|---|---|---|---|
| | Cases No. (95% UI) | ASR per 100,000 No. (95% UI) | Cases No. (95% UI) | ASR per 100,000 No. (95% UI) | EAPC (95% CI) |
| HL | 182,007 (78,811−252,106) | 16.78 (7.24−23.16) | 74,191 (46,299−100,603) | 4.13 (2.63−5.62) | −4.98 (−5.22 to −4.74) |
| NHL | 959,489 (821,882−1,166,903) | 92.42 (79.19−112.44) | 1,277,097 (997,263−1,551,800) | 66.95 (52.49−80.23) | −1.08 (−1.33 to −0.84) |
| BL | 16,103 (7496−23,364) | 1.44 (0.67−2.09) | 9516 (4826−13,346) | 0.68 (0.35−0.95) | −3.67 (−4.19 to −3.16) |
| Other NHL | 943,386 (813,448−1,150,563) | 90.98 (78.40−110.97) | 1,267,581 (989,522−1,538,728) | 66.27 (52.05−79.32) | −1.05 (−1.30 to −0.80) |
| MM | 46,854 (32,056−92,593) | 5.02 (3.42−9.99) | 338,359 (213,669−447,635) | 16.12 (10.09−21.35) | 3.15 (2.47−3.84) |
| MD/MP & other HN | 73,410 (48,256−124,011) | 7.50 (4.95−12.69) | 184,690 (128,102−288,696) | 9.77 (6.77−15.59) | 1.01 (0.91−1.11) |

ALL: acute lymphoid leukemia; AML: acute myeloid leukemia, CLL: chronic lymphoid leukemia; CML: chronic myeloid leukemia; HL: Hodgkin lymphoma; BL: Burkitt lymphoma; NHL: non-Hodgkin lymphoma; MM: multiple myeloma; MD/MP & other HN: myelodysplastic, myeloproliferative, and other hematopoietic neoplasms; ASR: age-standardized rates; EAPC: estimated annual percentage change.

43,626−74,039) and 3.42 (95% UI: 2.51−4.26) per 100,000 individuals, respectively, reflecting a significant decrease in mortality compared to 1990, when the ASMR was 6.46 (95% UI: 5.04−7.66) per 100,000 persons (Table 1). The DALY cases of leukemia totaled 3,924,466 (95% UI: 2,969,934−4,726,963) with an ASDR of 343.57 (95% UI: 260.79−414.78) per 100,000 individuals in 1990, which decreased to 2,205,221 (95% UI: 1,612,839−2,736,625) with an ASDR of 151.54 (95% UI: 108.71−185.06) per 100,000 persons in 2021 (Table 1). From 1990 to 2021, both the ASMR and ASDR of different leukemia subtypes decreased, with a significant decrease in CML and a minor decrease in CLL (Table 1). In contrast, the ASPR, ASIR, ASMR, and ASDR of leukemia in Taiwan increased from 1990 to 2021. CML is the only disease type with reduced mortality rates. The ASMR of CML was 0.37 (95% UI: 0.22−0.58) per 100,000 persons in 1990, decreasing to 0.21 (95% UI: 0.18−0.25) per 100,000 individuals in 2021. Similarly, the DALYs of CML decreased from 12.55 (95% UI: 7.47−19.92) per 100,000 persons in 1990 to 6.25 (95% UI: 5.29−7.45) per 100,000 individuals in 2021 (Table 2).

In 1990, the total number of HL in mainland China was 12,935 (95% UI: 5607−17,906), resulting in an ASPR of 1.16 (95% UI: 0.50−1.61) per 100,000 persons. This number increased to 23,818 (95% UI: 14,287−31,455) by 2021, with a prevalence rate of 1.36 (95% UI: 0.83−1.79) per 100,000 individuals (Table 1). However, the ASIR, ASMR, and ASDR of HL decreased from 1990 to 2021. The EAPC were −2.73 (95% CI: −3.00 to −2.47) for incidence, −4.71 (95% CI: −4.94 to −4.49) for mortality, and −4.98 (95% CI: −5.22 to −4.74) for disability (Table 1). The prevalence of NHL increased significantly, rising from 80,302 (95% UI: 68,643–95,538) in 1990–613,152 (95% UI: 476,112–762,582) in 2021. In addition, the incident cases rose from 31,216 (95% UI: 26,893−37,951) to 110,924 (95% UI: 86,934−135,200). The ASPR and ASIR of NHL increased, with the ASPR rising from 7.19 (95% UI: 6.19−8.55) to 31.17 (95% UI: 24.42−38.29) per 100,000 persons and the ASIR increasing from 3.32 (95% UI: 2.86−4.03) to 5.53 (95% UI: 4.36−6.68) per 100,000 individuals (Table 1). Patients with NHL and its subsets exhibited decreasing trends in mortality and DALYs. The most pronounced declines were observed in BL with an EAPC of ASMR at −2.52 (95% CI: −2.99 to −2.06) and ASDR at −3.67 (95% CI: −4.19 to −3.16) (Table 1). In Taiwan, the disease burden of HL was comparable to that in mainland China; however, a stable ASPR for HL was recorded (Table 2). Furthermore, NHL showed a slight decrease in ASDR in Taiwan, dropping from 110.72 (95% UI: 105.82−115.72) to 99.44 (95% UI: 89.79−109.53) per 100,000 individuals. However, there was no improvement in the NHL mortality rate (Table 2).

In mainland China, the number of prevalent, incident, deceased, and DALY cases of MM increased sharply from 2977 (95% UI: 2052−5851), 1693 (95% UI: 1154−3360), 1591 (95% UI: 1080−3159), and 46,854 (95% UI: 32,056−92,593) in 1990, to 47,004 (95% UI: 29,544−62,136), 17,250 (95% UI: 11,017−22,663), 12,984 (95% UI: 8448−17,114), and 338,359 (95% UI: 213,669−447,635) in 2021. MM demonstrated the greatest increase in ASPR with an EAPC to 5.96 (95% CI:

**Table 2. Cases and age-standardized rates of hematological malignancies in Taiwan and their temporal trends.**

| | 1990 | | 2021 | | 1990-2021 |
|---|---|---|---|---|---|
| | Cases No. (95% UI) | ASR per 100,000 No. (95% UI) | Cases No. (95% UI) | ASR per 100,000 No. (95% UI) | EAPC (95% CI) |
| **Prevalence** | | | | | |
| Leukemia | 1409 (1257−1563) | 7.53 (6.69−8.37) | 4724 (4180−5314) | 18.21 (15.56−21.07) | 3.33 (2.97−3.69) |
| AML | 427 (298−570) | 2.31 (1.59−3.15) | 931 (757−1132) | 7.57 (5.68−9.93) | 4.69 (4.21−5.17) |
| ALL | 421 (352−507) | 2.13 (1.79−2.55) | 999 (878−1129) | 3.35 (2.91−3.88) | 1.56 (1.30−1.81) |
| CLL | 225 (173−291) | 1.26 (0.97−1.64) | 1683 (1418−2007) | 4.16 (3.52−4.96) | 4.00 (3.60−4.41) |
| CML | 104 (61−170) | 0.56 (0.33−0.89) | 329 (267−405) | 0.95 (0.78−1.17) | 2.16 (1.72−2.61) |
| Other leukemia | 232 (191−280) | 1.28 (1.06−1.55) | 783 (610−954) | 2.17 (1.70−2.63) | 2.30 (1.93−2.67) |
| HL | 501 (434−575) | 2.41 (2.11−2.76) | 877 (740−1032) | 2.93 (2.42−3.59) | 0.03 (−0.39−0.45) |
| NHL | 3626 (3357−3907) | 18.14 (16.7−19.59) | 14,805 (12,933−16,806) | 41.76 (36.68−47.32) | 2.33 (1.83−2.84) |
| BL | 81 (56−124) | 0.42 (0.29−0.65) | 249 (98−455) | 0.89 (0.34−1.52) | 3.07 (2.58−3.55) |
| Other NHL | 3545 (3273−3823) | 17.72 (16.27−19.19) | 14,556 (12,698−16,551) | 40.87 (35.97−46.18) | 2.32 (1.81−2.83) |
| MM | 478 (421−533) | 2.78 (2.45−3.09) | 2681 (2281−3117) | 6.55 (5.50−7.64) | 2.89 (2.56−3.21) |
| MD/MP & other HN | 6216 (4944−7580) | 37.04 (29.62−45.24) | 16,133 (13,792−19,007) | 40.5 (34.71−47.46) | 0.30 (0.29−0.31) |
| **Incidence** | | | | | |
| Leukemia | 595 (567−623) | 3.30 (3.14−3.46) | 1655 (1496−1799) | 5.15 (4.67−5.64) | 1.67 (1.40−1.95) |
| AML | 137 (101−172) | 0.71 (0.52−0.90) | 211 (182−241) | 1.22 (0.98−1.51) | 2.42 (2.07−2.77) |
| ALL | 300 (262−351) | 1.67 (1.47−1.97) | 922 (835−1004) | 2.57 (2.34−2.78) | 1.40 (1.11−1.70) |
| CLL | 38 (30−50) | 0.23 (0.18−0.31) | 242 (205−288) | 0.59 (0.50−0.70) | 3.17 (2.79−3.55) |
| CML | 80 (48−126) | 0.46 (0.27−0.71) | 158 (129−195) | 0.44 (0.36−0.53) | 0.11 (−0.19−0.41) |
| Other leukemia | 39 (32−48) | 0.23 (0.19−0.28) | 122 (95−149) | 0.33 (0.25−0.40) | 1.69 (1.33−2.05) |
| HL | 75 (65−85) | 0.37 (0.33−0.42) | 117 (99−138) | 0.38 (0.32−0.47) | −0.49 (−0.85 to −0.13) |
| NHL | 840 (795−887) | 4.76 (4.50−5.01) | 2651 (2330−3005) | 7.00 (6.16−7.87) | 0.86 (0.52−1.21) |
| BL | 11 (8−18) | 0.06 (0.04−0.09) | 38 (15−69) | 0.13 (0.05−0.22) | 3.04 (2.56−3.51) |
| Other NHL | 829 (785−874) | 4.70 (4.44−4.96) | 2614 (2289−2958) | 6.87 (6.01−7.73) | 0.83 (0.48−1.17) |
| MM | 174 (158−189) | 1.06 (0.97−1.16) | 825 (725−936) | 1.98 (1.75−2.24) | 2.08 (1.76−2.41) |
| MD/MP & other HN | 430 (353−507) | 2.39 (1.97−2.83) | 1010 (864−1157) | 2.58 (2.24−2.92) | 0.27 (0.26−0.28) |
| **Mortality** | | | | | |
| Leukemia | 490 (468−511) | 2.82 (2.68−2.94) | 1197 (1091−1291) | 3.28 (3.01−3.53) | 0.63 (0.38−0.88) |
| AML | 102 (76−125) | 0.53 (0.39−0.65) | 129 (116−142) | 0.44 (0.40−0.49) | −0.13 (−0.38−0.11) |
| ALL | 276 (242−322) | 1.59 (1.40−1.86) | 813 (733−880) | 2.20 (2.01−2.38) | 1.07 (0.77−1.36) |
| CLL | 18 (14−23) | 0.13 (0.10−0.16) | 70 (58−84) | 0.17 (0.14−0.20) | 1.12 (0.76−1.48) |
| CML | 61 (36−96) | 0.37 (0.22−0.58) | 82 (68−100) | 0.21 (0.18−0.25) | −1.72 (−1.93 to −1.5) |
| Other leukemia | 34 (28−41) | 0.21 (0.17−0.25) | 103 (79−123) | 0.25 (0.20−0.30) | 1.16 (0.77−1.54) |
| HL | 19 (17−21) | 0.11 (0.10−0.12) | 16 (14−19) | 0.04 (0.04−0.05) | −3.51 (−3.78 to −3.24) |
| NHL | 578 (552−606) | 3.49 (3.32−3.67) | 1471 (1313−1630) | 3.62 (3.26−4.02) | −0.22 (−0.45−0.02) |
| BL | 4 (3−6) | 0.02 (0.02−0.03) | 6 (2−10) | 0.02 (0.01−0.03) | 0.20 (−0.06−0.46) |
| Other NHL | 574 (547−602) | 3.47 (3.30−3.65) | 1465 (1304−1623) | 3.60 (3.23−3.99) | −0.22 (−0.46−0.02) |
| MM | 127 (118−136) | 0.80 (0.75−0.86) | 544 (490−598) | 1.29 (1.16−1.42) | 1.57 (1.23−1.90) |
| MD/MP & other HN | 49 (28−78) | 0.33 (0.19−0.52) | 170 (147−194) | 0.42 (0.36−0.47) | 0.60 (0.38−0.83) |
| **DALYs** | | | | | |
| Leukemia | 22,306 (21,369−23,274) | 113.87 (109.08−118.68) | 33,791 (30,980−36,420) | 114.16 (105.12−123.11) | 0.21 (0.04−0.37) |
| AML | 5884 (4375−7363) | 28.84 (21.42−36.09) | 4874 (4408−5403) | 21.96 (19.87−24.84) | −0.38 (−0.66 to −0.1) |
| ALL | 12,238 (10,554−14,276) | 62.75 (54.27−73.27) | 22,399 (20,671−24,153) | 74.22 (68.48−79.76) | 0.60 (0.43−0.77) |
| CLL | 547 (431−710) | 3.22 (2.54−4.15) | 1804 (1509−2143) | 4.61 (3.87−5.44) | 1.26 (1.06−1.45) |
| CML | 2428 (1426−3888) | 12.55 (7.47−19.92) | 2112 (1765−2555) | 6.25 (5.29−7.45) | −2.07 (−2.22 to −1.93) |
| Other leukemia | 1210 (994−1445) | 6.51 (5.39−7.87) | 2602 (2057−3103) | 7.13 (5.66−8.48) | 0.78 (0.50−1.06) |
| HL | 827 (736−915) | 4.12 (3.68−4.54) | 552 (474−645) | 1.70 (1.44−2.04) | −3.41 (−3.66 to −3.15) |

*(Continued)*

**Table 2.** (Continued)

| | 1990 | | 2021 | | 1990-2021 |
|---|---|---|---|---|---|
| | Cases No. (95% UI) | ASR per 100,000 No. (95% UI) | Cases No. (95% UI) | ASR per 100,000 No. (95% UI) | EAPC (95% CI) |
| NHL | 20,666 (19,828 − 21,519) | 110.72 (105.82 − 115.72) | 36,310 (32,557 − 40,098) | 99.44 (89.79 − 109.53) | −0.68 (−0.88 to −0.48) |
| BL | 213 (149 − 331) | 1.08 (0.75 − 1.67) | 223 (89 − 385) | 0.97 (0.38 − 1.58) | 0.26 (−0.07 − 0.58) |
| Other NHL | 20,453 (19,617 − 21,312) | 109.64 (104.74 − 114.61) | 36,086 (32,415 − 39,814) | 98.47 (89.06 − 108.42) | −0.69 (−0.89 to −0.48) |
| MM | 3617 (3362 − 3867) | 20.96 (19.49 − 22.39) | 13,269 (12,023 − 14,507) | 32.8 (29.82 − 35.68) | 1.44 (1.14 − 1.73) |
| MD/MP & other HN | 1774 (1090 − 2686) | 10.26 (6.28 − 15.50) | 4357 (3765 − 4960) | 11.74 (10.19 − 13.26) | 0.35 (0.14 − 0.55) |

ALL: acute lymphoid leukemia; AML: acute myeloid leukemia, CLL: chronic lymphoid leukemia; CML: chronic myeloid leukemia; HL: Hodgkin lymphoma; BL: Burkitt lymphoma; NHL: non-Hodgkin lymphoma; MM: multiple myeloma; MD/MP & other HN: myelodysplastic, myeloproliferative, and other hematopoietic neoplasms; ASR: age-standardized rates; EAPC: estimated annual percentage change.

5.35 − 6.56), ASIR with an EAPC of 4.05 (95% CI: 3.38 − 4.73), ASMR with an EAPC of 3.11 (95% CI: 2.40 − 3.83), and ASDR with an EAPC of 3.15 (95% CI: 2.47 − 3.84) 1990–2021 (Table 1). Similarly, the increasing patterns of MM recorded in Taiwan mirror those recorded in mainland China (Table 2). Additionally, the prevalence, incidence, mortality, and DALYs of MD/MP and other hematological neoplasms showed increasing trends in mainland China and Taiwan (Tables 1 and 2).

### Disease burden of HMs by age and gender

In 2021, HMs was more common in males and was predominantly reported among older adults in terms of prevalence, incidence, mortality, and DALYs (Figs 1–2). ALL mainly affects children, particularly those aged < 5 years who have the highest ASPR and ASIR values. However, ALL mortality was higher in individuals aged > 60 years. This led to two peaks in the ALL DALYs across different age groups (Fig 1A). Although AML can occur in all age groups, the ASPR and ASDR are typically higher among older adults. Nonetheless, AML incidence and mortality were more common in individuals aged 60 and older, although a decline was observed in those aged 95 and above (Fig 1B). The prevalence, incidence, and DALYs of CLL and CML were primarily found in individuals over 20 years of age, with rates increasing with age. Nevertheless, a significant decline in ASPR was noted among people aged over 80 years, particularly in those aged 95 years and older. Additionally, chronic leukemia mortality primarily affected individuals aged > 60 years (Fig 1C–D). Likewise, the ASPR, ASIR, ASMR, and ASDR rates for other types of leukemia increased with age in individuals aged 20 and older; however, declines were observed in the 85 − 89 and 95 + age groups (Fig 1E).

The prevalence, incidence, mortality, and DALYs of HL were recorded for all age groups. Significantly higher rates were observed in individuals aged > 50 years, whereas a declining trend was noted in those aged > 90 years (Fig 2A). The disease burden of BL primarily affects individuals aged 60 years. However, relatively high ASDR rates were observed in patients aged < 20 years (Fig 2B). The ASPR of other NHL cases increased with age, peaking in the 70 − 74 age group and declining thereafter. NHL primarily occurred in individuals over 30 years of age, whereas the mortality rate was predominantly observed in those over 50 years of age (Fig 2C).

MM primarily affects elderly adults, with prevalence rates that first increase and then decrease with age. The ASIR and ASMR of MM showed a similar change pattern, peaking in the 85–90-year age group. Conversely, the ASDR of MM fluctuated more with age, showing two peaks in the 70–74 years and 85–90 years age groups (Fig 2D). The ASPR, ASMR, and ASDR of MD/MPs and other hematological neoplasms similarly increased with age in older adults. Additionally, a high ASIR was noted in the 40 − 79 age group (Fig 2E). The age and gender-specific distributions of ASPR, ASIR, ASMR, and ASDR of hematological neoplasms in Taiwan generally resemble those found in mainland China (S1 and S2 Fig).

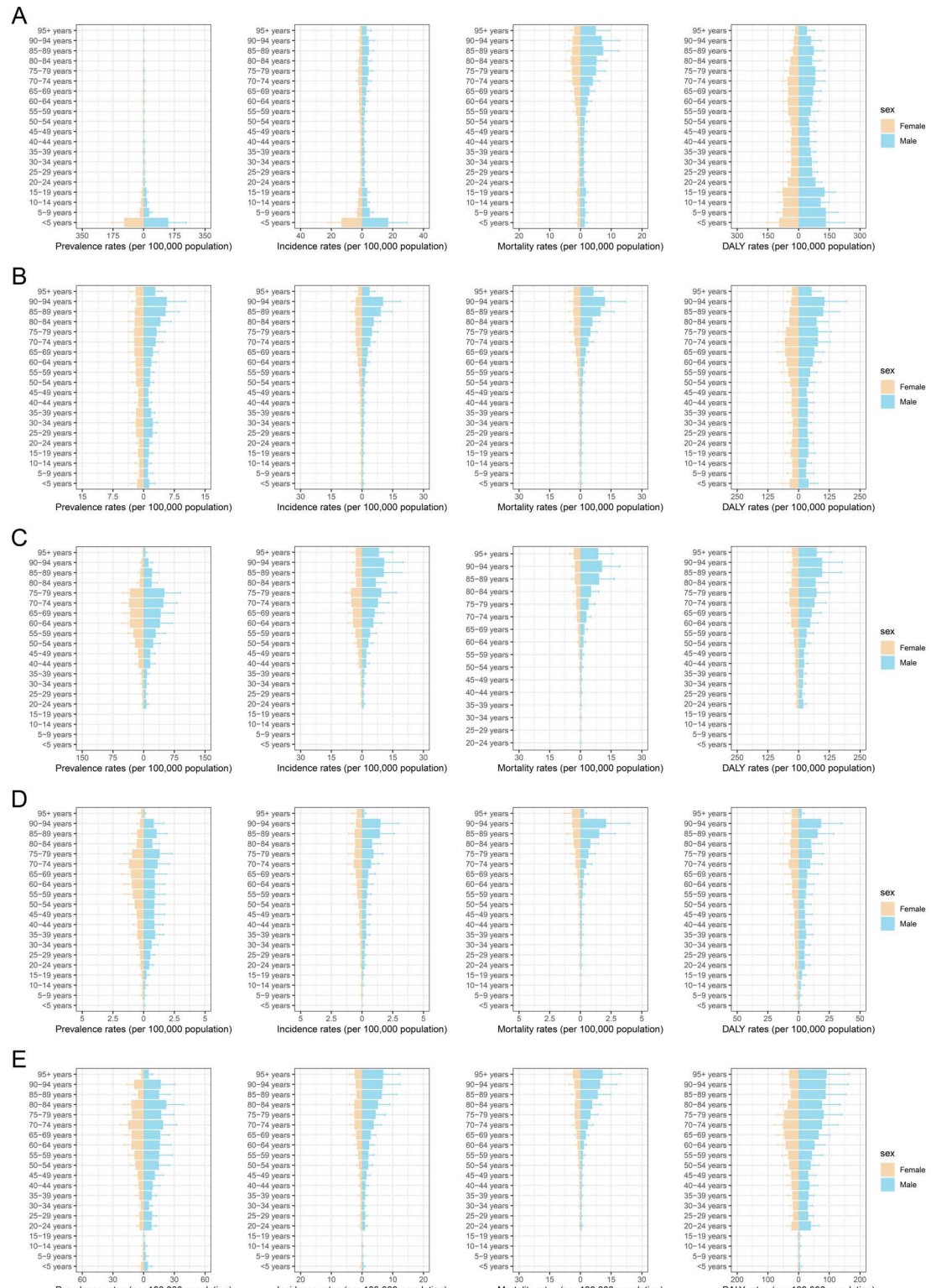

**Fig 1. Age- and sex-specific distribution of prevalence, incidence, mortality, and DALY rates for leukemia in mainland China. (A)** Distributions of age-standardized prevalence rates (ASPR), incidence rates (ASIR), mortality rates (ASMR), and DALY rates (ASDR) for acute lymphoid leukemia (ALL). **(B)** Distributions of ASPR, ASIR, ASMR, ASDR for acute myeloid leukemia (AML). **(C)** Distributions of ASPR, ASIR, ASMR, ASDR for chronic lymphoid leukemia (CLL). **(D)** Distributions of ASPR, ASIR, ASMR, ASDR for chronic myeloid leukemia (CML). **(E)** Distributions of ASPR, ASIR, ASMR, ASDR for other leukemia.

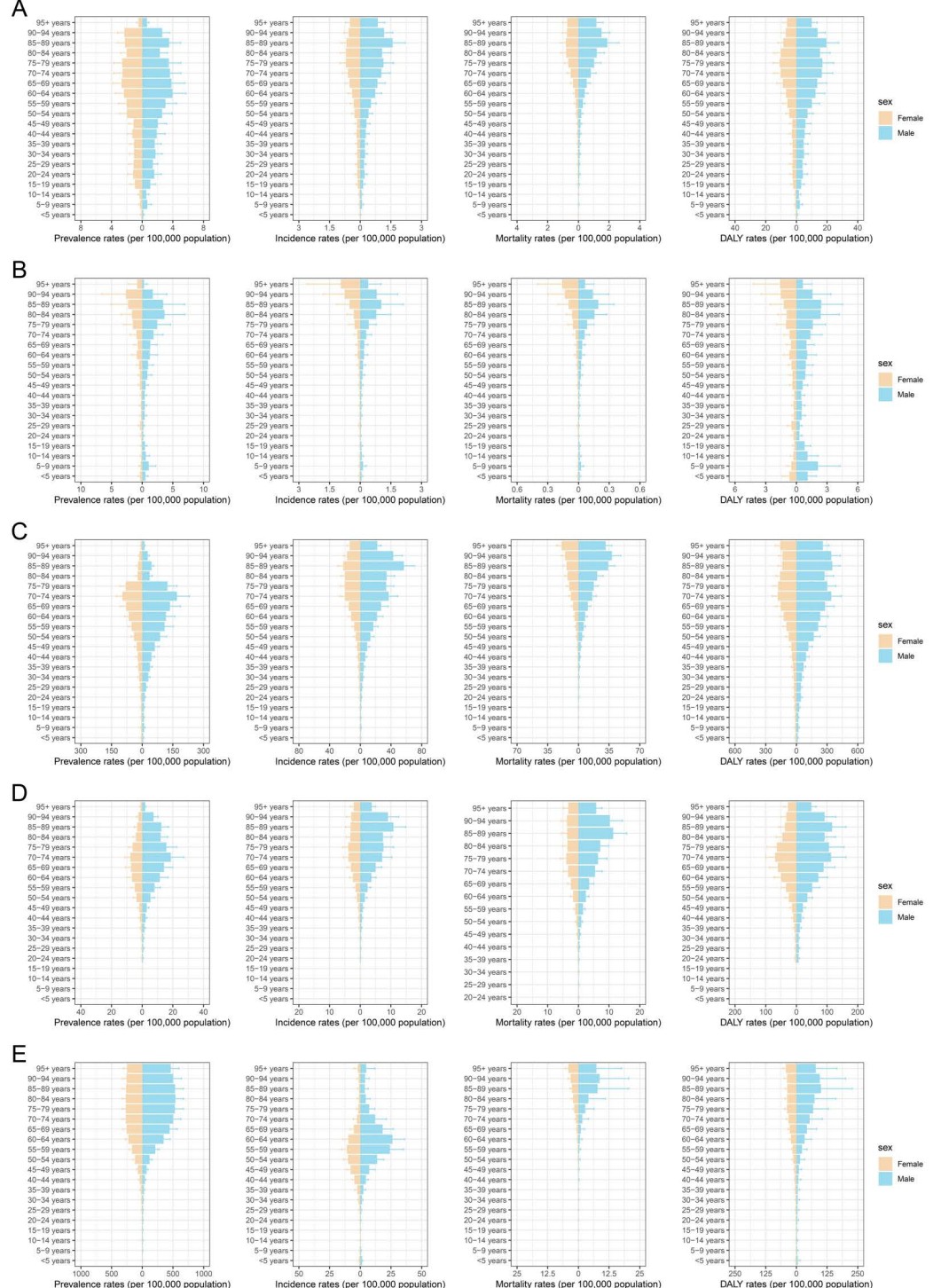

**Fig 2. Age- and sex-specific distribution of prevalence, incidence, mortality, and DALY rates for lymphoma, multiple myeloma, and other hematological neoplasms in mainland China. (A)** Distributions of age-standardized prevalence rates (ASPR), incidence rates (ASIR), mortality rates (ASMR), and DALY rates (ASDR) for Hodgkin lymphoma (HL). **(B)** Distributions of ASPR, ASIR, ASMR, ASDR for Burkitt lymphoma (BL). **(C)** Distributions of ASPR, ASIR, ASMR, ASDR for other non-Hodgkin lymphoma (NHL). **(D)** Distributions of ASPR, ASIR, ASMR, ASDR for multiple myeloma (MM). **(E)** Distributions of ASPR, ASIR, ASMR, ASDR for myelodysplastic, myeloproliferative (MD/MP), and other hematopoietic neoplasms.

## Overall temporal trends of disease burden for HMs by gender

In mainland China, the prevalence and incidence of lymphoid leukemia increased from 1990 to 2021, particularly among males. Recently, there has been a decline in the ASPR and ASIR for ALL. During the same period, the ASMR and ASDR of lymphoid leukemia decreased, especially among females and patients with ALL (S3 Fig). The ASPR, ASIR, ASMR, and ASDR of AML patients decreased from 1990 to 2021 (S3 Fig). In males, CML showed an increase in the ASPR, whereas the rates in females remained stable. However, ASIR, ASMR, and ASDR of the CML decreased during this period (S3 Fig). Males had a higher prevalence, incidence, mortality, and DALYs for other leukemias than did females. However, both sexes experienced slight declines in ASIR, ASMR, and ASDR (S3 Fig). Although the ASPR of HL increased, the incidence, mortality, and DALYs decreased significantly, particularly in males (S4 Fig). Males experienced a significantly larger increase in ASPR and ASIR in BL and other NHL. In contrast, males displayed an overall declining trend in ASMR and ASDR, although some fluctuations were observed (S4 Fig). Between 1990 and 2021, the prevalence, incidence, mortality, and DALYs of MM significantly increased, with a notable male predisposition (S4 Fig). Men showed a more pronounced increase in ASPR and ASIR for MD/MP and other HMs than did women. Moreover, there was an overall decrease in ASMR and ASDR, although increases were observed during specific periods (S4 Fig). S5–6 Fig depict the trends in disease burden associated with HMs in Taiwan.

## Temporal joinpoint analysis of disease burden for HMs

Joinpoint regression analysis revealed a significant increase in the overall ASPR for ALL in mainland China from 1990 to 2021 (AAPC = 3.87%; 95% CI: 3.49%–4.26%; $P < 0.0001$). The most notable increase occurred from 2003 to 2008 (APC = 10.73%; 95% CI: 9.88% to 11.59%; $P < 0.0001$). However, a decreased trajectory was observed from 2019 to 2021 (APC = −6.58%; 95% CI: −10.59% to −2.39%; $P = 0.0048$) (Fig 3A and S1 Table). The ASIR of ALL decreased at first, then rose, and finally fell again, remaining in a totally stable status (AAPC = 0.15%; 95% CI: −0.18% to 40.47%; $P = 0.3817$) (Fig 3A and S2 Table). A global downward trend was documented for ASMR (AAPC = −2.56%; 95% CI: −2.78% to −2.34%; $P < 0.0001$) and ASDR of ALL (AAPC = −3.12%; 95% CI: −3.39% to −2.84%; $P < 0.0001$) with a significant decline occurring from 2000 to 2006 (APC = −4.26%; 95% CI: −4.68% to −3.84%; $P < 0.0001$) and from 1999 to 2006 (APC = −5.29%; 95% CI: −5.73% to −4.83%; $P < 0.0001$), respectively (Fig 3A and S3−4 Table). Overall downward trends were noted for ASPR (AAPC = −1.06%; 95% CI: −1.35% to −0.77%; $P < 0.0001$), ASIR (AAPC = −1.10%; 95% CI: −1.22% to −0.97%; $P < 0.0001$), ASMR (AAPC = −1.59%; 95% CI: −1.74% to −1.44%; $P < 0.0001$), and ASDR (AAPC = −2.27%; 95% CI: −2.53% to −2.01%; $P < 0.0001$) for AML from 1990 to 2021, with sharp reductions identified during 2004−2007 for ASPR (APC = −4.24%; 95% CI: −5.98% to −2.47%; $P = 0.0001$), ASMR (APC = −4.49%; 95% CI: −5.39% to −3.59%; $P < 0.0001$), and ASDR (APC = −5.84%; 95% CI: −7.40% to −4.26%; $P < 0.0001$) (Fig 3A and S1−4 Table). The ASPR (AAPC = 4.30%; 95% CI: 4.04% to 4.56%; $P < 0.0001$) and ASIR (AAPC = 2.25%; 95% CI: 2.15% to 2.35%; $P < 0.0001$) of CLL increased, while the ASMR (AAPC = −0.72%; 95% CI: −0.91% to −0.53%; $P < 0.0001$) and ASDR (AAPC = −0.60%; 95% CI: −0.74% to −0.46%; $P < 0.0001$) dropped, with significant declines occurring from 2004 to 2007 for ASMR (APC = −3.43%; 95% CI: −4.75% to −2.09%; $P < 0.0001$) and ASDR (APC = −3.52%; 95% CI: −4.86% to −2.17%; $P < 0.0001$) (Fig 3A and S1−4). Although the ASPR (AAPC = 0.85%; 95% CI: 0.59% to 1.10%; $P < 0.0001$) of CML increased, the ASIR (AAPC = −1.84%; 95% CI: −2.24% to −1.44%; $P < 0.0001$), ASMR (AAPC = −4.04%; 95% CI: −4.30% to −3.78%; $P < 0.0001$), and ASDR (AAPC = −4.62%; 95% CI: −4.91% to −4.32%; $P < 0.0001$) all showed an downward trend from 1990 to 2021 (Fig 3A and S1−4 Table). The most significant increase in ASPR of CML occurred during 2016–2021 (APC = 3.88%; 95% CI: 2.92% to 4.86%; $P < 0.0001$), while the utmost reduction of ASIR (APC = −5.82%; 95% CI: −8.40% to −3.17%; $P = 0.0003$), ASMR (APC = −9.79%; 95% CI: −11.13% to −8.43%; $P < 0.0001$), and ASDR (APC = −10.68%; 95% CI: −12.32% to −9.01%; $P < 0.0001$) were recorded during 2004–2007 (Fig 3A and S1 − 4 Table). The trends of disease burden for other leukemia mirrored those of CML except for a stale ASPR (AAPC = 0.04%; 95% CI: −0.12% to 0.19%; $P = 0.6614$) (Fig 3A and S1−4 Table). An overall upward trend was observed in ASPR for HL (AAPC = 0.48%; 95% CI: 0.17% to 0.80%; $P = 0.0028$),

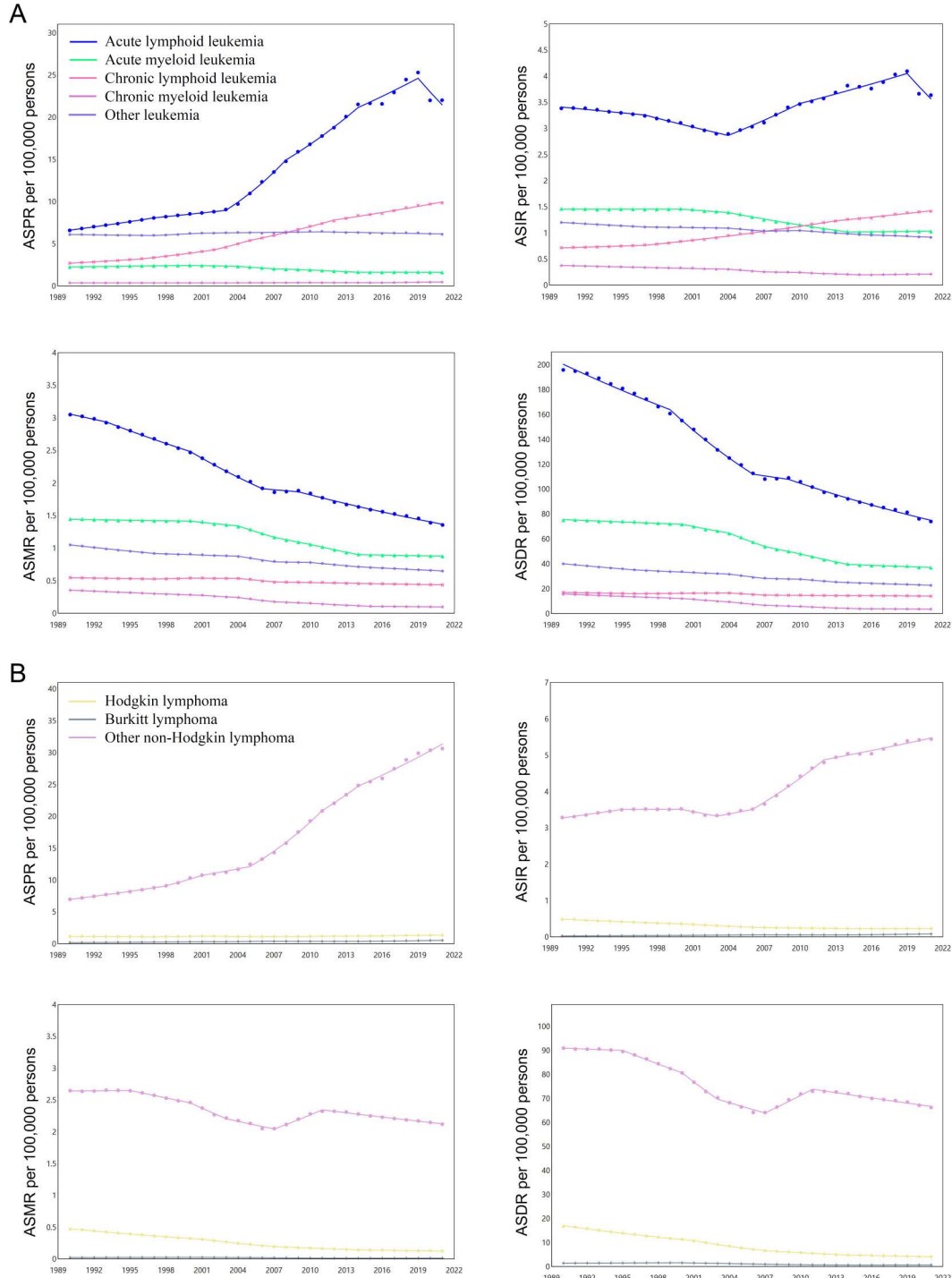

**Fig 3. Disease burden trends of leukemia and lymphoma analyzed by joinpoint regression analysis in mainland China.** Overall trends of age-standardized prevalence rates (ASPR), incidence rates (ASIR), mortality rates (ASMR), and DALY rates (ASDR) for leukemia **(A)** and lymphoma **(B)** from 1990 to 2021, as indicated.

BL (AAPC = 3.37%; 95% CI: 2.74% to 4.01%; *P* < 0.0001), and other NHL (AAPC = 4.96%; 95% CI: 4.51% to 5.42%; *P* < 0.0001) (Fig 3B and S1−4 Table). The ASIR of HL (APC = −2.39%; 95% CI: −2.48% to −2.31%; *P* < 0.0001) showed a diminishing trend, while that of BL (APC = 3.57%; 95% CI: 3.27% to 3.87%; *P* < 0.0001) and other NHL (APC = 1.67%; 95% CI: 1.39% to 1.96%; *P* < 0.0001) significantly increased from 1990 to 2021 (Fig 3B and S1−4 Table). Both ASMR and ASDR of lymphoma decreased during this period, with HL being the most significant subtype (AAPC of ASMR = −4.23% with 95% CI: −4.34% to −4.11%; AAPC of ASDR = −4.48% with 95% CI: −4.60% to −4.37%; both *P* values were < 0.0001) (Fig 3B and S1−4 Table). The ASPR (AAPC = 6.43%; 95% CI: 5.90% to 6.96%; *P* < 0.0001), ASIR (AAPC = 4.74%; 95% CI: 4.18% to 5.29%; *P* < 0.0001), ASMR (AAPC = 3.91%; 95% CI: 2.79% to 5.05%; *P* < 0.0001), and ASDR (AAPC = 3.89%; 95% CI: 3.31% to 4.46%; *P* < 0.0001) of MM all exhibited an upward trend from 1990 to 2021, with the greatest increase observed during 1992–1995 (Fig 4A and S1−4 Table). Global increases in ASPR (AAPC = 0.36%; 95% CI: 0.34% to 0.39%; *P* < 0.0001), ASIR (AAPC = 0.60%; 95% CI: 0.58% to 0.63%; *P* < 0.0001), ASMR (AAPC = 1.52%; 95% CI: 1.45% to 1.59%; *P* < 0.0001), and ASDR (AAPC = 0.84%; 95% CI: 0.81% to 0.88%; *P* < 0.0001) of MD/MP and other hematological neoplasms were recorded during this period (Fig 4B and S1−4 Table). The trends in disease burden changes for HMs in Taiwan are presented in S7–8 Fig and S5–8 Tables.

### Projected future disease burden of HMs

The BAPC model was used to predict the impact of blood cancer in China over the next decade. In mainland China, the incidence of acute leukemia is expected to decline from 2022 to 2032, especially among females with ALL and males with AML (Fig 5A−B and S9 Table). Conversely, the incidence of chronic leukemia was projected to increase or remain stable (Fig 5C–D and S9 Table). Additionally, the incidence rates of other leukemias were anticipated to decrease in both sexes (Fig 5E and S9 Table). Changes in the ASIR of the NHL group are expected to be more pronounced than those of the HL group, especially for the BL group, which is projected to show rising rates. Intriguingly, alterations in other NHL were expected to display opposite trends between males and females (Fig 5F–H and S9 Table). The incidence of MM is projected to increase slightly, whereas that of MD/MP and other hematological neoplasms is expected to remain comparable (Fig 5I–J and S9 Table). The mortality rates of most HMs are expected to decline from 2022 to 2032, excluding BL, MM, MD/MP, and other hematological neoplasms that are projected to display rising or stable rates (Fig 6A–J and S9 Table). In Taiwan, the incidence rates of myeloid and other leukemias are predicted to decrease, whereas that of lymphoid leukemia remains stable or shows a slight increase (S9 Fig and S10 Table). Following this trend, the ASIR of lymphoma was expected to decrease from 2022 to 2032 (S9 Fig and S10 Table). The incidence of MM was predicted to remain stable in both males and females (S9 Fig and S10 Table). The ASIR of MD/MP and other hematological neoplasms were expected to increase slightly during this period (S9 Fig and S10 Table). With the exception of BL, MM, CLL, MD/MP, and other hematological neoplasms, the ASMR of other blood cancers is projected to decline from 2022 to 2032 in Taiwan (S10 Fig and S10 Table).

### Discussion

HMs represent a heterogeneous group of cancers originating from hematopoietic and lymphoid tissues and are characterized by distinct biological behaviors and widely varying prognoses. Despite significant progress in the development of novel diagnostic techniques, such as next-generation sequencing for genetic profiling and the introduction of targeted therapies, immunotherapies, and stem cell transplantation, they remain a formidable challenge to global public health. Recent reports have highlighted the persistently high morbidity and mortality rates associated with these diseases, underscoring the urgent need for a comprehensive understanding of their epidemiological patterns [22,31,32]. In China, the burden of HMs has not yet been fully elucidated. Given the significant regional disparities in healthcare access, lifestyle factors, and genetic predispositions between mainland China and Taiwan, a comprehensive analysis of the disease burden across these regions is essential. Here, we characterized temporal trends in the incidence, prevalence, mortality,

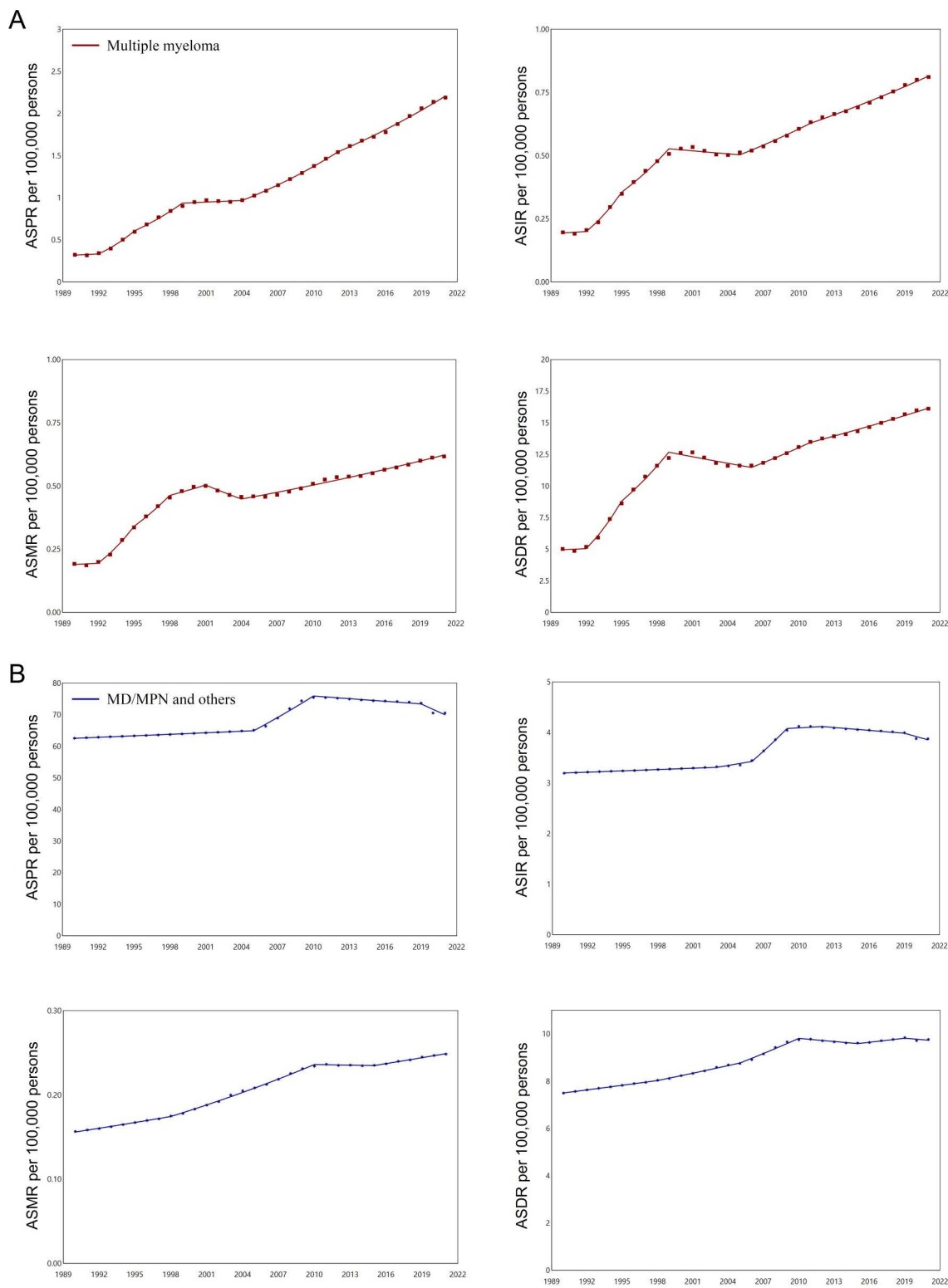

**Fig 4. Disease burden trends of multiple myeloma and other hematological neoplasms analyzed by joinpoint regression analysis in mainland China.** Overall trends of age-standardized prevalence rates (ASPR), incidence rates (ASIR), mortality rates (ASMR), and DALY rates (ASDR) for multiple myeloma (MM) **(A)** and myelodysplastic, myeloproliferative (MD/MP), and other hematopoietic neoplasms **(B)** from 1990 to 2021, as indicated.

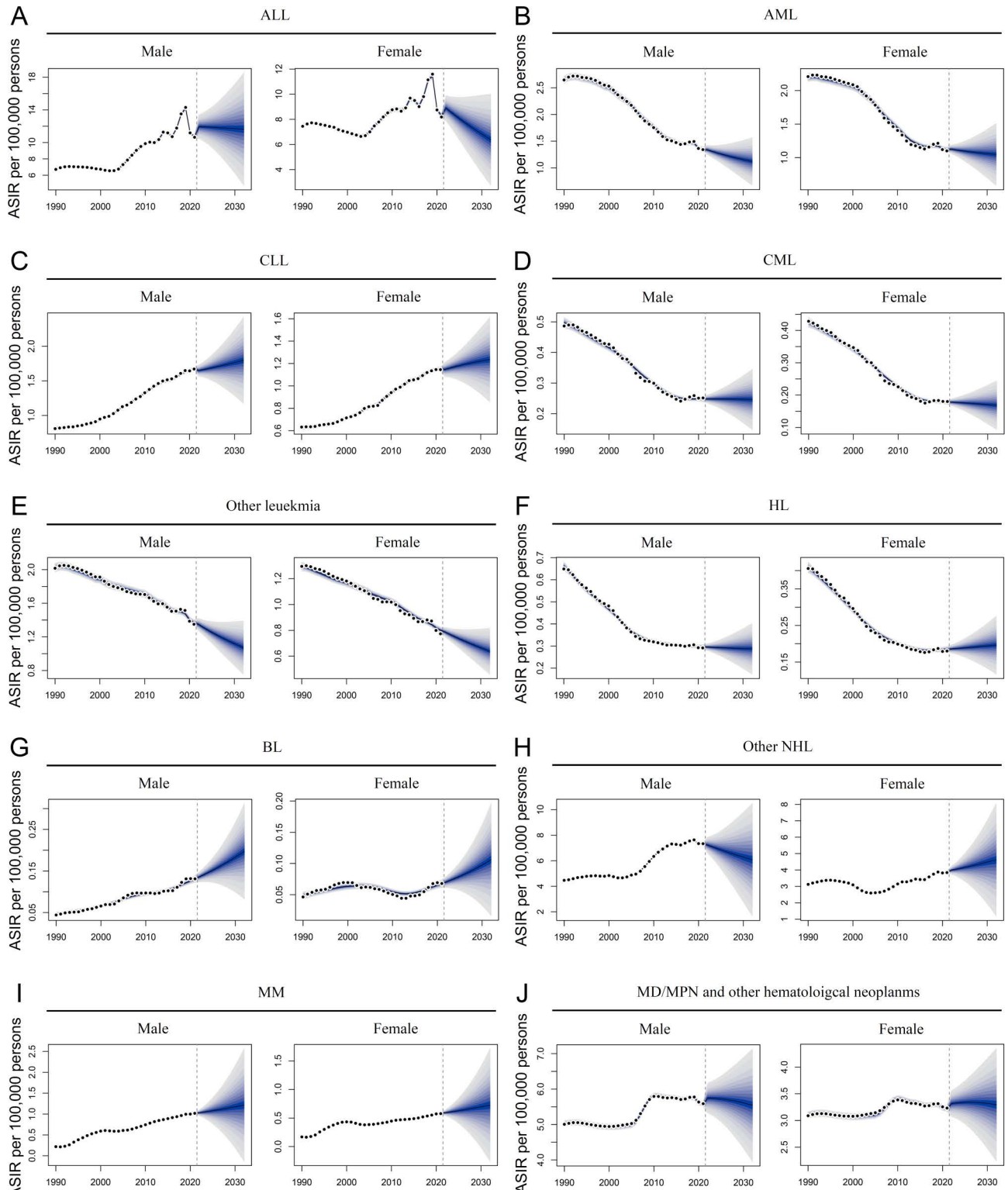

**Fig 5. Projections of incidence for hematological malignancies in mainland China.** Overall trends of age-standardized incidence rates (ASIR) for acute lymphoid leukemia (ALL) **(A)**, acute myeloid leukemia (AML) **(B)**, chronic lymphoid leukemia (CLL) **(C)**, chronic myeloid leukemia **(D)**, other leukemia **(E)**, Hodgkin lymphoma (HL) **(F)**, Burkitt lymphoma (BL) **(G)**, other non-Hodgkin lymphoma (NHL) **(H)**, multiple myeloma (MM) **(I)**, and myelodysplastic, myeloproliferative (MD/MP), and other hematopoietic neoplasms **(J)** from 1990 to 2032, as indicated.

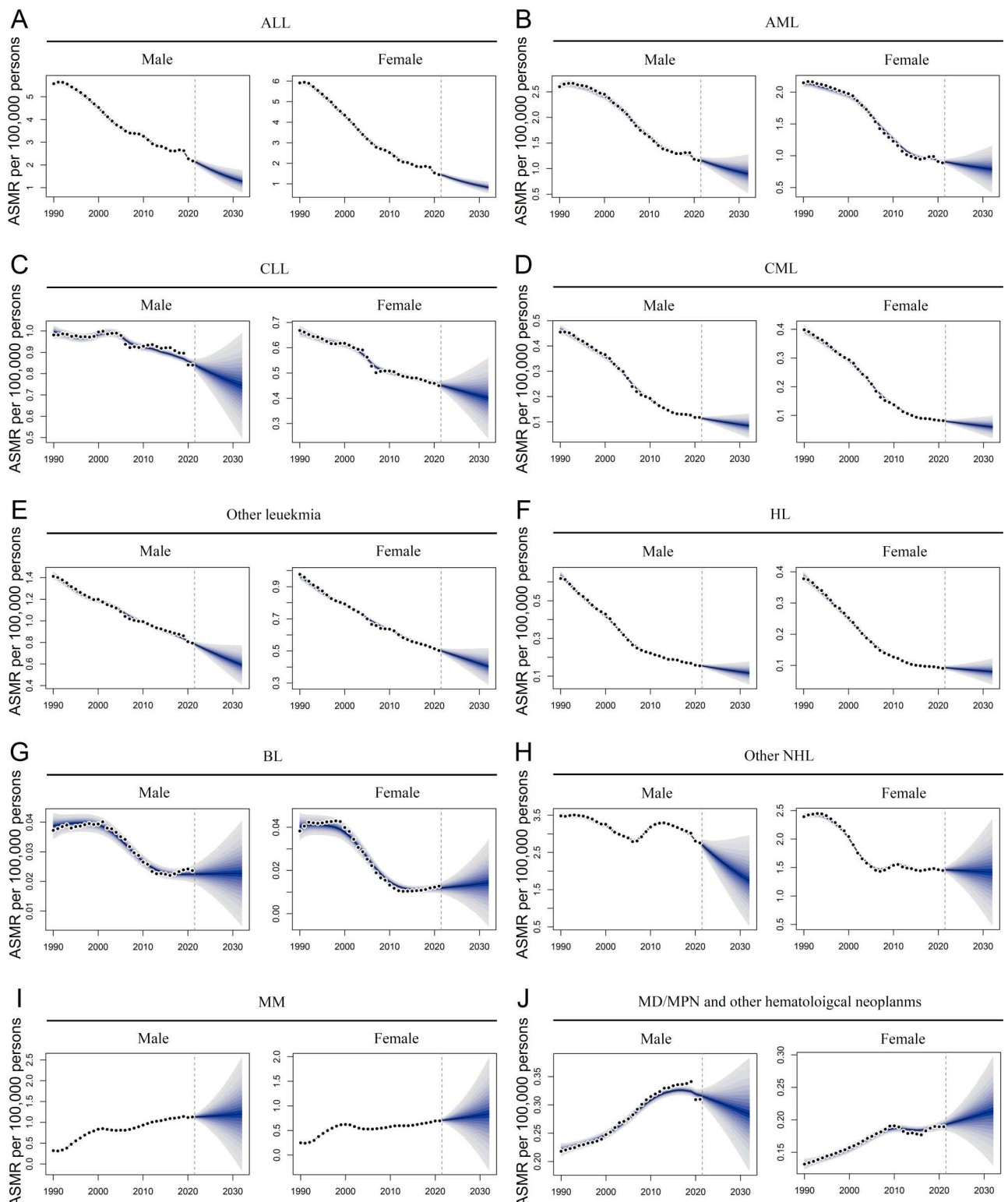

**Fig 6. Projections of mortality for hematological malignancies in mainland China.** Overall trends of age-standardized mortality rates (ASMR) for acute lymphoid leukemia (ALL) **(A)**, acute myeloid leukemia (AML) **(B)**, chronic lymphoid leukemia (CLL) **(C)**, chronic myeloid leukemia **(D)**, other leukemia **(E)**, Hodgkin lymphoma (HL) **(F)**, Burkitt lymphoma (BL) **(G)**, other non-Hodgkin lymphoma (NHL) **(H)**, multiple myeloma (MM) **(I)**, and myelodysplastic, myeloproliferative (MD/MP), and other hematopoietic neoplasms **(J)** from 1990 to 2032, as indicated.

and DALYs of HMs from 1990 to 2021. By integrating historical data with future projections, our study offers a holistic view of the current and impending challenges associated with HMs, which is important for guiding evidence-based healthcare policies, resource allocation, and research prioritization.

Between 1990 and 2021, mainland China saw an increase in the prevalence and incidence of leukemia and lymphoma, whereas mortality and DALYs decreased. This is consistent with nationwide statistics on cancer incidence and mortality in mainland China (from 31 provinces) recently released by the National Cancer Center of China [33]. However, in Taiwan, leukemia showed an increasing trend in prevalence, incidence, mortality, and DALYs during the same period, which is supported by studies from the Taiwan Cancer Registry [34,35]. Conversely, lymphoma mortality and DALYs decreased from 1990 to 2021. Additionally, MM, MD/MP, and other hematopoietic neoplasms demonstrated significant increases in ASPR, ASIR, ASMR, and ASDR in mainland China and Taiwan, which is consistent with the global epidemiological landscape of these diseases [36]. Although these results indicate improvements in the diagnosis, treatment strategies, and prognoses of patients with HMs in China, such as the use of anti-CD20 antibodies in patients with lymphoma and the development of new prognostic stratification systems during this period [37,38], Taiwan Province continues to face greater challenges in this area. Notably, MM imposes a significant disease burden on the Chinese population, particularly in mainland China, which reported the greatest increase in prevalence, incidence, mortality, and DALYs among hematological neoplasms.

In 2021, lymphoid leukemia, which includes ALL and CLL, was the most prevalent type of leukemia in mainland China. However, AML was the only leukemic form that showed a decline in both ASPR and ASIR, suggesting a reduction in new cases. Increased rates of prevalence and decreased incidence were observed for CML and HL in mainland China between 1990 and 2021, reflecting an improvement in the outcomes of these two types of hematological cancers. The underlying elements reflected in these trends are advances in treatment techniques and the emergence of new drugs. Imatinib was approved in mainland China in 2002 and has been used to treat CML patients with a high rate of cytogenetic and hematological responses in those who failed previous interferon therapy [39]. Thereafter, it has been widely used in clinical practice, leading to significant improvements in the prognosis of patients with CML; the same situation has occurred in Taiwan [40]. Intriguingly, ALL, CLL, NHL, and MM showed increases in ASPR and ASIR during this period. In contrast, all hematological cancers exhibited rising rates, except for HL, in Taiwan between 1990 and 2021. Generally, the disease burden of myeloid leukemia (AML and CML) decreased, whereas that of lymphoid neoplasms (leukemia, lymphoma, and MM) increased in mainland China, which was not observed in Taiwan during this period. The BAPC model predicted that the incidence of substantial lymphoid neoplasms is expected to increase over the next decade. Thus, while Taiwan faces a more complicated disease burden of HMs, mainland China should pay particular attention to lymphoid malignancies. Moreover, joinpoint analysis demonstrated that the most significant increase in the prevalence and incidence of lymphoid neoplasms occurred in the first 10 years of this century in mainland China. In 2001, China formally entered the World Trade Organization, which had a profound impact on environmental factors, industrial structure, economic development, population expansion and migration, trade patterns, and lifestyle in China, which may have contributed to the landscape changes of HMs in mainland China [41–43]. In a population-based case-control study that included both myeloid and lymphoid neoplasms, hair dye use for at least 15 years was associated with a higher risk of lymphoid malignancies among females [44]. The excess cancer risk associated with 1,3-butadiene inhalation exposure was assessed using an extensive dataset from a previous study. Additionally, the excess risk in the general population is highest for lymphoid neoplasms [45]. Chronic inflammation and infection are more prevalent in lymphoid malignancies than in myeloid neoplasms [46]. Moreover, chronic inflammation and exposure to pesticides or organic solvents are risk factors for MM [47]. Therefore, lifestyle choices, occupational exposure to specific substances, and chronic infections increase the risk of lymphoid neoplasms. Additionally, the Healthy China 2000 Goals enhance China's healthcare system, potentially leading to more diagnoses of these diseases. Collectively, these factors may affect the disease burden of lymphoid cancer in China.

In mainland China, the ASMR and ASDR of most hematological neoplasms have been declining. Joinpoint analysis showed that the most significant decline in mortality rates occurred during the first 10 years of this century, which is consistent with the trends in prevalence and incidence rates. This may reflect the heightened influence of new international medical advances on Chinese patients, as well as the targeted health policies implemented in China. Additionally, the rapid advancement of haploidentical hematopoietic stem cell transplantation in mainland China has contributed to a reduction in mortality from blood cancers. This is especially true for patients with high-risk or relapsed diseases, as well as for those without siblings donors [48–50]. Clinical outcomes for patients in Taiwan require improvement compared to those in mainland China, particularly as mortality and DALYs are increasing for AML, CLL, and other leukemias. This finding suggests that clinicians and health policymakers should intensify their efforts to control these diseases in Taiwan. Notably, the mortality rates of MM, MD/MP, and other hematological neoplasms increased significantly from 1990 to 2021 in both mainland China and Taiwan. Furthermore, the BAPC model indicated that the ASMR of MM and females with MD/MP and other hematological neoplasms is expected to increase over the next ten years. This highlights the need for an increased focus on these diseases.

This study has limitations that are common to other GBD estimation initiatives. First, large-scale infectious disease epidemics, such as severe acute respiratory syndrome and coronavirus disease 2019, may have disrupted clinical workflows for data collection and patient management in HMs. Second, the GBD study data primarily relied on reported sources rather than direct submissions, potentially introducing issues of data completeness, timeliness, and quality. Despite efforts to standardize the GBD modeling frameworks, these limitations persist and may affect the accuracy and cross-study comparability of the results. Additionally, the lack of granular data precludes a detailed analysis of the epidemiological characteristics of these patient populations. Finally, caution is warranted when interpreting the data from Taiwan because of its relatively small patient population and healthcare system differences from mainland China, which may lead to substantial variability in the findings.

## Conclusion

This study systematically analyzed the disease burden of HMs in mainland China and Taiwan from 1990 to 2021 using the GBD Study 2021 data with decadal projections. Key findings revealed divergent trends between regions: mainland China saw increased prevalence/incidence of leukemia (particularly AML and CLL) and NHL but declining mortality/ DALYs, reflecting improved treatments. Conversely, Taiwan experienced overall increases in leukemia burden (prevalence, incidence, mortality, and DALYs), although decreased trends could be observed in certain subtypes of leukemia and NHL. MM and other hematological neoplasms demonstrated significant upward trends in both regions. Lymphoid neoplasms drive mainland China's burden, which is likely linked to environmental and lifestyle factors, whereas MM emerges as the fastest-growing subtype. Projections indicate a declining acute leukemia incidence in mainland China but stable/increasing chronic leukemia/NHL (notably BL). In contrast, Taiwan faces stable lymphoid leukemia, declining myeloid leukemia, and rising MM/BL mortality. The implications include prioritizing early detection of lymphoid malignancies/MM in mainland China, improving leukemia treatments in Taiwan, enhancing surveillance for environmental risks, expanding geriatric oncology services, and adopting stem cell transplantation. Future research should investigate regional risk factor disparities, refine projection models with clinical data, foster cross-regional collaboration, and explore patient-centered outcomes to inform precise public health strategies and reduce the total burden of HMs.

## Supporting information

**S1 Fig. Age- and sex-specific distribution of prevalence, incidence, mortality, and DALY rates for leukemia in Taiwan province.**
(DOCX)

**S2 Fig.  Age- and sex-specific distribution of prevalence, incidence, mortality, and DALY rates for lymphoma, multiple myeloma, and other hematological neoplasms in Taiwan province.**
(DOCX)

**S3 Fig.  Temporal trends of disease burden for leukemia in mainland China by sex (1990−2021).**
(DOCX)

**S4 Fig.  Temporal trends of disease burden for lymphoma, multiple myeloma, and other hematological neoplasms in mainland China by sex (1990−2021).**
(DOCX)

**S5 Fig.  Temporal trends of disease burden for leukemia in Taiwan province by sex (1990−2021).**
(DOCX)

**S6 Fig.  Temporal trends of disease burden for lymphoma, multiple myeloma, and other hematological neoplasms in Taiwan Province by sex (1990−2021).**
(DOCX)

**S7 Fig.  Disease burden trends of leukemia and lymphoma analyzed by joinpoint regression analysis in Taiwan province.**
(DOCX)

**S8 Fig.  Disease burden trends of multiple myeloma and other hematological neoplasms analyzed by joinpoint regression analysis in Taiwan province.**
(DOCX)

**S9 Fig.  Projections of incidence for hematological malignancies in Taiwan province.**
(DOCX)

**S10 Fig.  Projections of mortality for hematological malignancies in Taiwan province.**
(DOCX)

**S1 Table.  Temporal joinpoint analysis of ASPR for hematological malignancies in mainland China (1990−2021).**
(DOCX)

**S2 Table.  Temporal joinpoint analysis of ASIR for hematological malignancies in mainland China (1990−2021).**
(DOCX)

**S3 Table.  Temporal joinpoint analysis of ASMR for hematological malignancies in mainland China (1990−2021).**
(DOCX)

**S4 Table.  Temporal joinpoint analysis of ASDR for hematological malignancies in mainland China (1990−2021).**
(DOCX)

**S5 Table.  Temporal joinpoint analysis of ASPR for hematological malignancies in Taiwan (1990−2021).**
(DOCX)

**S6 Table.  Temporal joinpoint analysis of ASIR for hematological malignancies in Taiwan (1990−2021).**
(DOCX)

**S7 Table.  Temporal joinpoint analysis of ASMR for hematological malignancies in Taiwan (1990−2021).**
(DOCX)

**S8 Table. Temporal joinpoint analysis of ASDR for hematological malignancies in Taiwan (1990−2021).**
(DOCX)

**S9 Table. Predicted incidence and mortality of hematological malignancies in mainland China.**
(DOCX)

**S10 Table  Predicted incidence and mortality rates of hematological malignancies in Taiwan province.**
(DOCX)

**S1 Data  The raw data used in this study.**
(ZIP)

## Acknowledgments

We deeply appreciate the contributions of the GBD Study 2021 collaborators. We would like to extend our sincere gratitude to the staff of the Department of Hematology and the Department of Clinical Epidemiology at the First Hospital of Jilin University for their invaluable support throughout this study.

## Author contributions

**Conceptualization:** Su-Jun Gao, Long Su.

**Data curation:** Yu Fu, Ya-Zhe Du, Yun-Wei Zhang, Fei Song, Long Su.

**Formal analysis:** Yu Fu, Ya-Zhe Du, Yun-Wei Zhang, Fei Song, Su-Jun Gao, Long Su.

**Funding acquisition:** Long Su.

**Investigation:** Yu Fu, Ya-Zhe Du, Yun-Wei Zhang, Fei Song, Su-Jun Gao, Long Su.

**Methodology:** Yu Fu, Ya-Zhe Du, Yun-Wei Zhang, Fei Song, Su-Jun Gao, Long Su.

**Project administration:** Yu Fu, Su-Jun Gao, Long Su.

**Resources:** Yu Fu, Ya-Zhe Du, Fei Song, Su-Jun Gao, Long Su.

**Software:** Yu Fu, Ya-Zhe Du, Long Su.

**Supervision:** Su-Jun Gao, Long Su.

**Validation:** Yu Fu, Ya-Zhe Du, Yun-Wei Zhang, Fei Song, Su-Jun Gao, Long Su.

**Visualization:** Yu Fu, Yun-Wei Zhang, Fei Song, Long Su.

**Writing – original draft:** Yu Fu, Ya-Zhe Du, Su-Jun Gao, Long Su.

**Writing – review & editing:** Yu Fu, Ya-Zhe Du, Yun-Wei Zhang, Fei Song, Su-Jun Gao, Long Su.

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
