## [Decision Letter · Decision Letter 0]

PONE-D-25-02174Disease burden of hematological malignancies in China from 1990 to 2021 and forecasts for the next decade: a systemic analysis of the global burden of disease study 2021PLOS ONE

Dear Dr. Su,

Thank you for submitting your manuscript to PLOS ONE. After careful consideration, we feel that it has merit but does not fully meet PLOS ONE’s publication criteria as it currently stands. Therefore, we invite you to submit a revised version of the manuscript that addresses the points raised during the review process.

The study highlights the insights of hematological malignancies with a glimpse of epidemiological data and a glimpse into the future. One of the points that is misleading is how these trajectories could lead to therapeutic approaches, which is lacking in the manuscript, although it is in the abstract or cover letter. Moreover, the introduction part is so long that I could not get the idea or the rationale behind. If we consider this as a story narrative, the introduction part should prepare the reader in a more eye-catching way.

We look forward to receiving your revised manuscript.

Kind regards,

Mehmet Baysal

Academic Editor

PLOS ONE

Journal Requirements:

2. Please note that funding information should not appear in the Acknowledgments section or other areas of your manuscript. We will only publish funding information present in the Funding Statement section of the online submission form. Please remove any funding-related text from the manuscript.

[This work was supported by the Norman Bethune Program of Jilin University [grant number 2022B17, 2022]; Talent Reserve Program (TRP), the First Hospital of Jilin University [grant number JDYYCB-2023007, 2023].].

[The authors declare no conflict of interest.].

5. Please include captions for your Supporting Information files at the end of your manuscript, and update any in-text citations to match accordingly. Please see our Supporting Information guidelines for more information: http://journals.plos.org/plosone/s/supporting-information .

Reviewers' comments:

Reviewer's Responses to Questions

**Comments to the Author**

1. Is the manuscript technically sound, and do the data support the conclusions?

Reviewer #1: Yes

2. Has the statistical analysis been performed appropriately and rigorously? 

Reviewer #1: I Don't Know

3. Have the authors made all data underlying the findings in their manuscript fully available?

Reviewer #1: No

4. Is the manuscript presented in an intelligible fashion and written in standard English?

Reviewer #1: Yes

5. Review Comments to the Author

Reviewer #1: I have carefully read and evaluated the manuscript; Here are my suggestions and concerns

The paper does not specify from where the site/database the dataset was downloaded and how the authors acquired this access. This information is crucial for reproducibility. Give details about sources of data, including restrictions or permissions required.

Title and abstract fail to highlight crucial features of the study, such as the utilization of Taiwan-specific data or long-term estimates. Consider revision for highlighting these key features.

Obvious grammatical/typographical errors pervade the manuscript. A careful edit of language for clarity and professionalism is recommended.

The methodology section must be more described in detail. For example:

How were the variables chosen or verified?

Were there any exclusion rules?

Describe the reason why certain analytical strategies (e.g., projection models) were adopted.

The paper does not have a separate conclusion summarizing main findings, implications, and future research directions. The conclusion section should be included to make the paper more impactful.

Discussion needs to be lengthened to:

Provide better contextualization of results within the existing literature.

Discuss limitations (e.g., data level of detail, regional biases such as the inclusion of Taiwan).

Do not add new data/analyses not presented in the results.

If a systematic review is done, than it must be done with the adherance to the PRISMA guidelines like presenting a flow diagram as well as detailed reporting on search strategy, screening, and eligibility criteria.

6. PLOS authors have the option to publish the peer review history of their article (what does this mean? ). If published, this will include your full peer review and any attached files.

**Do you want your identity to be public for this peer review?** For information about this choice, including consent withdrawal, please see our Privacy Policy .

Reviewer #1: No

---

## [Author Response · Author response to Decision Letter 1]

30 May 2025

Point-by-point response to comments from the editor and reviewer

We are grateful to the the editor and reviewer for their helpful and constructive comments, and have revised the manuscript in response to these comments. The following is a point-by-point response to their comments (all changes in the revised manuscript are marked in red color).

Editor #1: Comments to the Author

The study highlights the insights of hematological malignancies with a glimpse of epidemiological data and a glimpse into the future. One of the points that is misleading is how these trajectories could lead to therapeutic approaches, which is lacking in the manuscript, although it is in the abstract or cover letter. Moreover, the introduction part is so long that I could not get the idea or the rationale behind. If we consider this as a story narrative, the introduction part should prepare the reader in a more eye-catching way.

Reply: We appreciate your feedback on the introduction. To address the concerns, we’ve strengthened the link between disease burden trends and therapeutic implication, such as highlighting how China’s younger diagnostic ages and molecular markers inform targeted therapy strategies. We also streamlined the narrative to enhance flow, starting with global challenges, then China’s uniqueness, and finally our study’s objectives to ensure clarity and relevance. These revisions better connect epidemiology with clinical implications in a shorter format, making the rationale more compelling.

Reviewer #1: Comments to the Author

I have carefully read and evaluated the manuscript. Here are my suggestions and concerns:

The paper does not specify from where the site/database the dataset was downloaded and how the authors acquired this access. This information is crucial for reproducibility. Give details about sources of data, including restrictions or permissions required.

Reply: We thank the reviewer for pointing out this issue and appreciate your feedback regarding the data source and access details, which are indeed critical for reproducibility. The dataset utilized in this study was derived from publicly available data in the GBD database, maintained by the Institute for Health Metrics and Evaluation (IHME). As stated in IHME’s Free-of-Charge Non-Commercial User Agreement, these aggregated public data are accessible without formal application for non-commercial academic use. We directly downloaded the data via the GBD platform (https://www.healthdata.org), which requires no special approval for such public datasets. To align with transparency standards, we have updated the manuscript’s Materials and Methods section to clarify: “Publicly available data on HMs in mainland China and Taiwan (a province of China) were derived from the 2021 GBD investigation on November 20th 2024 (https://www.healthdata.org), which provided a comprehensive evaluation of health impairments using the latest epidemiological data and refined methodologies. These datasets are freely usable for noncommercial research under the Institute for Health Metrics and Evaluation (IHME) (marked in red color)”.

2. Title and abstract fail to highlight crucial features of the study, such as the utilization of Taiwan-specific data or long-term estimates. Consider revision for highlighting these key features.

Reply: We thank the reviewer for pointing out this issue and have revised the title and abstract highlighting these crucial features of this study (marked in red color).

3. Obvious grammatical/typographical errors pervade the manuscript. A careful edit of language for clarity and professionalism is recommended.

Reply: We thank the reviewer for pointing out this issue. Our revised manuscript was edited by an international company for scientific language (www.editage.com), and the certification was provided as well.

4. The methodology section must be more described in detail. For example: How were the variables chosen or verified? Were there any exclusion rules? Describe the reason why certain analytical strategies (e.g., projection models) were adopted.

Reply: We thank the reviewer for pointing out this issue and have revised methodology section marked in red color. Age-standardized prevalence rates (ASPR), incidence rates (ASIR), mortality rates (ASMR), and DALY rates (ASDR) were selected for cancer epidemiological studies using the GBD database, as these metrics comprehensively characterize the disease burden across multiple dimensions. The ASPR captures the population disease burden at a specific time to inform resource allocation, and the ASIR identifies new cases to assess risk profiles and primary prevention effectiveness. The ASMR reflects disease severity and healthcare system impact, whereas the ASDR integrates mortality and disability to provide a holistic view of health losses. Together, these metrics align with the GBD’s global standardized framework for comparative analysis. The GBD study ensures data quality through rigorous standardization and statistical processing, including the selection of reliable data sources, dataset cleaning, application of uniform definitions, use of advanced modeling, and multilevel reviews by global collaborators. Given the preexisting data quality control processes for GBD, no additional exclusion criteria were applied in this study. Moreover, the reason why we choose certain analytical strategies were also added as the reviewer recommended.

5. The paper does not have a separate conclusion summarizing main findings, implications, and future research directions. The conclusion section should be included to make the paper more impactful.

Reply: We thank the reviewer for pointing out this issue and have added a dedicated conclusion section that succinctly summarizes our main findings, implications, and future research directions. Specifically, the conclusion highlights divergent hematological malignancy trends between mainland China and Taiwan, the rising burden of multiple myeloma, and age/gender disparities. It emphasizes public health priorities like enhanced surveillance for lymphoid neoplasms and geriatric care, alongside clinical strategies such as stem cell transplantation adoption. For future research, we outline plans to investigate regional risk factor disparities, refine projection models with clinical data, and foster cross-regional collaborations. This addition strengthens the manuscript’s impact by consolidating key insights and guiding next steps in reducing global disease burden.

6. Discussion needs to be lengthened to: provide better contextualization of results within the existing literature; discuss limitations (e.g., data level of detail, regional biases such as the inclusion of Taiwan); do not add new data/analyses not presented in the results.

Reply: We thank the reviewer for pointing out this issue. We have lengthened the discussion section by citing more existing literature, and discuss the limitattions. Moreover, no new data/analyses not presented in the results. All those changes were marked in red color in secdtion of Discussion.

---

## [Decision Letter · Decision Letter 1]

Hematological malignancy burden in mainland China and Taiwan from 1990 to 2021 and decadal projections: insights from the Global Burden of Disease Study 2021

PONE-D-25-02174R1

Dear Dr. Su,

We’re pleased to inform you that your manuscript has been judged scientifically suitable for publication and will be formally accepted for publication once it meets all outstanding technical requirements.

Kind regards,

Mehmet Baysal

Academic Editor

PLOS ONE

Additional Editor Comments (optional):

Reviewers' comments:

Reviewer's Responses to Questions

**Comments to the Author**

1. If the authors have adequately addressed your comments raised in a previous round of review and you feel that this manuscript is now acceptable for publication, you may indicate that here to bypass the “Comments to the Author” section, enter your conflict of interest statement in the “Confidential to Editor” section, and submit your "Accept" recommendation.

Reviewer #1: All comments have been addressed

2. Is the manuscript technically sound, and do the data support the conclusions?

Reviewer #1: Yes

3. Has the statistical analysis been performed appropriately and rigorously? 

Reviewer #1: Yes

4. Have the authors made all data underlying the findings in their manuscript fully available?

Reviewer #1: Yes

5. Is the manuscript presented in an intelligible fashion and written in standard English?

Reviewer #1: Yes

6. Review Comments to the Author

Reviewer #1: The authors correctly responded to all comments and addressed in the revised version. With this revised

version the manuscript has been evolved into a scientifically sound paper. I congratulate them for the

revision of their paper. I have no further request.

7. PLOS authors have the option to publish the peer review history of their article (what does this mean? ). If published, this will include your full peer review and any attached files.

**Do you want your identity to be public for this peer review?** For information about this choice, including consent withdrawal, please see our Privacy Policy .

Reviewer #1: No

---

## [Editor Report · Acceptance letter]

PONE-D-25-02174R1

PLOS ONE

Dear Dr. Su,

I'm pleased to inform you that your manuscript has been deemed suitable for publication in PLOS ONE. Congratulations! Your manuscript is now being handed over to our production team.

Kind regards,

on behalf of

Dr. Mehmet Baysal

Academic Editor

PLOS ONE